# Identification and Clinical Significance of Pancreatic Cancer Stem Cells and Their Chemotherapeutic Drug Resistance

**DOI:** 10.3390/ijms24087331

**Published:** 2023-04-15

**Authors:** Yu-Chi Kuo, Hao-Wei Kou, Chih-Po Hsu, Chih-Hong Lo, Tsann-Long Hwang

**Affiliations:** Department of Surgery, Chang Gung Memorial Hospital, Chang Gung University, Lin-Kou, Taoyuan 333, Taiwan

**Keywords:** pancreatic ductal adenocarcinoma, cancer stem cells, CD44, EpCAM, ABCG2 transporter, chemoresistance

## Abstract

Pancreatic cancer ranks in the 10th–11th position among cancers affecting men in Taiwan, besides being a rather difficult-to-treat disease. The overall 5-year survival rate of pancreatic cancer is only 5–10%, while that of resectable pancreatic cancer is still approximately 15–20%. Cancer stem cells possess intrinsic detoxifying mechanisms that allow them to survive against conventional therapy by developing multidrug resistance. This study was conducted to investigate how to overcome chemoresistance and its mechanisms in pancreatic cancer stem cells (CSCs) using gemcitabine-resistant pancreatic cancer cell lines. Pancreatic CSCs were identified from human pancreatic cancer lines. To determine whether CSCs possess a chemoresistant phenotype, the sensitivity of unselected tumor cells, sorted CSCs, and tumor spheroid cells to fluorouracil (5-FU), gemcitabine (GEM), and cisplatin was analyzed under stem cell conditions or differentiating conditions. Although the mechanisms underlying multidrug resistance in CSCs are poorly understood, ABC transporters such as ABCG2, ABCB1, and ABCC1 are believed to be responsible. Therefore, we measured the mRNA expression levels of ABCG2, ABCB1, and ABCC1 by real-time RT-PCR. Our results showed that no significant differences were found in the effects of different concentrations of gemcitabine on CSCs CD44+/EpCAM+ of various PDAC cell line cultures (BxPC-3, Capan-1, and PANC-1). There was also no difference between CSCs and non-CSCs. Gemcitabine-resistant cells exhibited distinct morphological changes, including a spindle-shaped morphology, the appearance of pseudopodia, and reduced adhesion characteristics of transformed fibroblasts. These cells were found to be more invasive and migratory, and showed increased vimentin expression and decreased E-cadherin expression. Immunofluorescence and immunoblotting experiments demonstrated increased nuclear localization of total β-catenin. These alterations are hallmarks of epithelial-to-mesenchymal transition (EMT). Resistant cells showed activation of the receptor protein tyrosine kinase c-Met and increased expression of the stem cell marker cluster of differentiation (CD) 24, CD44, and epithelial specific antigen (ESA). We concluded that the expression of the ABCG2 transporter protein was significantly higher in CD44+ and EpCAM+ CSCs of PDAC cell lines. Cancer stem-like cells exhibited chemoresistance. Gemcitabine-resistant pancreatic tumor cells were associated with EMT, a more aggressive and invasive phenotype of numerous solid tumors. Increased phosphorylation of c-Met may also be related to chemoresistance, and EMT and could be used as an attractive adjunctive chemotherapeutic target in pancreatic cancer.

## 1. Introduction

Pancreatic ductal adenocarcinoma (PDAC) is the fourth leading cause of cancer mortality in the United States, whereas in Taiwan, it ranks in the 10th–11th position among all cancers frequently associated with high cancer mortality. The overall 5-year survival rate is only 5–10% [1,2,3,4,5,6,7,8,9,10,11,12], while that of resectable tumors has been reported to be approximately 15–20% [13]. Genetic or molecular changes associated with PDAC have been investigated over several years; however, the majority of previous studies using tumor samples primarily focused on the analysis of tumor DNA. Recently, additional analyses of tumor mRNA have also been performed. These were aimed at the identification of gross genomic alternations, specific genes with mutations, or mRNA expression [14,15]. Such studies have provided only partial information about protein products of mutated or dysregulated genes. Our research group has also focused on PDAC using microarray and proteomics techniques over the past few years, wherein we identified phosphoglycerate kinase 1 as a novel potential biomarker for the diagnosis of PDAC; we are also continuing related studies [16,17,18].

Cancer stem cells (CSCs) were first identified and demonstrated by transplantation of a small population of leukemic cells harvested from patients into immunodeficient mice, which subsequently developed the same type of cancer as the patients [19]. The researchers found that a subset of cells within the tumor possess stem cell-like characteristics such as the ability to initiate tumors, high proliferative rates, high capacity of self-renewal, and the propensity to differentiate into actively proliferating tumor cells [20,21,22]. In CSCs, the pathways of self-renewal and differentiation are generally deregulated, which results in unlimited self-renewal and a subsequent excess of CSCs. In addition, CSCs have aberrant differentiation programs that generate progenitor tumor cells, which then proliferate to form the bulk of the tumor [23]. Stem-like tumor cells are generally associated with increased expression of stem cell surface markers such as CD44 and CD133 [24,25].

Putative CSCs have been identified in pancreatic cancer based on the expression of the three surface markers CD44, CD24, and epithelial specific antigen (ESA) [26,27,28,29,30,31,32,33]. A subpopulation of pancreatic cancer cells showing CD44+CD24+ESA^+^ have been proven to possess stem cell-like properties of self-renewal and the ability to produce differentiated progeny. CSCs have intrinsic detoxifying mechanisms and can easily escape conventional treatments. Moreover, there has been increasing accumulating evidence explaining drug resistance based on CSC models in recent years. Such evidence suggests that CSC-targeted therapy is mandatory for overcoming drug resistance and curing tumors [34,35,36]. It has been reported that the expression of CD44 variants significantly correlates with poor prognosis in patients with colon or pancreatic cancer [37,38,39,40]. On the other hand, CD133 has been found to be a highly conserved antigen and a human homologue of mouse prominin-1. CD133 has been detected in several normal tissues, including neuroepithelium and embryonic and adult immature epithelia. Furthermore, studies have reported that CD133 mRNA expression is an independent prognostic factor for overall survival, based on increased expression levels detected in the peripheral blood of patients with cancer with bone metastasis [41,42,43].

The aim of a CSC-targeted therapy should be to overcome drug resistance for curing the tumor. PDAC is generally resistant to chemotherapeutic agents, which further indicates that it is important to overcome the mechanisms underlying drug resistance. CSCs are naturally resistant to chemotherapy due to their properties of quiescence, capacity for DNA repair, and ABC transporter expression [44,45,46]. Drug resistance to chemotherapy frequently occurs in cancers, especially PDAC, and is a major obstacle to successful cancer treatment. Studies using tumor cell lines have demonstrated that multidrug resistance (MDR) can develop and cause chemotherapy failure. Advances in elucidating the molecular basis of the MDR phenotype have indicated that elevated membrane expression of drug efflux pumps such as P-glycoprotein (Pgp or ABCB1), multidrug resistance protein 1 (MRP1 or ABCC1), and ABCG2 is a frequent cause of MDR in human cancers [47].

These drug efflux pumps belong to the superfamily of ATP-binding cassette (ABC) transporters with a wide variety of substrates. These ABC transporters share a common structural feature comprising transmembrane domains and cytoplasmic nucleotide-binding domains with walker motifs. The nucleotide-binding domains appear to function as an engine, providing the energy required for transport activities by hydrolyzing ATP. In humans, there are 49 members in this ABC transporter superfamily, which is divided into seven subfamilies (ABCA, ABCB, ABCC, ABCD, ABCE, ABCF, and ABCG). Several of these ABC transporter members are known to efflux anticancer drugs and thereby cause drug resistance when overexpressed in model cancer cell lines [48,49,50,51,52,53].

As drug resistance was always the problem during chemotherapy for patients with PDAC, this study was designed and conducted to investigate how to overcome chemoresistance and its mechanisms in pancreatic cancer stem cells (CSCs) using gemcitabine-resistant pancreatic cancer cell lines.

## 2. Results

### 2.1. Cytotoxicity of Gemcitabine to PDAC Cells and CSCs

The CSCs CD44+/CD44− of BxPC-3 PDAC cells that were treated with or without gemcitabine (25 nM, 72 h) showed a better survival rate in the CD44+ group than in the CD44− group (Figure 1, *p* < 0.05). The CSCs EpCAM+/EpCAM− of BxPC-3 PDAC cells that were treated with or without gemcitabine (25 nM, 72 h) also showed a better survival rate in EpCAM+ than in EpCAM− (Figure 2, *p* < 0.05).

### 2.2. Effects of Cytotoxicity of Verapamil on PDAC Cells and CSCs

IC50 values for gemcitabine in respective BxPC-3 CSCs cell fractions are shown in Table 1. When the survival of parental BxPC-3 PDAC cells treated with different concentrations of verapamil and gemcitabine was analyzed, it was found that 25 and 50 µM concentrations of verapamil exerted the best chemotherapeutic effects with a gemcitabine concentration of 0.001 µg/mL (Figure 3, *p* < 0.05). The survival of the CSCs CD44+/CD44− of BxPC-3 PDAC cells treated with different concentrations of verapamil and gemcitabine was also assessed, which revealed that 25 and 50 µM concentrations of verapamil exerted the best chemotherapeutic effects with a gemcitabine concentration of 0.001 µg/mL (Figure 4, *p* < 0.05).

### 2.3. Responsibility of ABC Transporters for the Acquisition of Multidrug Resistance

ABC transporters with expression altered by >1.5 fold in CD44+ cells compared with CD44− cells (RQ: Relative Quantity RNA) (Table 2). The expression levels of chemoresistance genes ABCC1, ABCG2, NOTCH2 and JAG-1 in BxPC-3 cells were compared between CSCs and non-CSCs, and only ABCG2 exhibited a higher expression in CD44+ CSCs than in CD44− non-CSCs. For EpCAM+ CSCs and EpCAM− non-CSCs, the difference was similar to that between CD44+ CSCs and CD44− non-CSCs (Figure 5 and Figure 6, *p* < 0.05).

The expression levels of ABCG2, NOTCH2 and JAG1 were slightly elevated in CSCs (CD44+, EpCAM+) than in non-CSCs after 48 h of treatment with 25 nM gemcitabine. The ABCG2 showed a significantly higher expression in CSCs than in non-CSCs (Figure 7 and Figure 8, *p* < 0.05).

## 3. Discussion

PDAC has an extremely poor prognosis because it generally invades the surrounding tissues and metastasizes to lymph nodes, liver, or peritoneum at the time of diagnosis. CD44 is an adhesion molecule and a membrane receptor for hyaluronan, and is involved in cell motility and metastases. The gene encoding CD44 generates a variety of isoforms by alternative splicing, which predominantly affects the proximal structure of the extracellular membrane of CD44 proteins. In other tumors, multiple cell surface markers may be used to purify CSCs, and CD44 has been used as one of the CSC markers.

CD44 is a member of the transmembrane glycoprotein family exhibiting a large number of isoforms identified in several human tissues, and shows a particularly high expression in proliferating cells and squamous cell epithelium. The CD44 variant 6 (v6) molecule has been identified as a marker for tumor metastasis and prognosis in several tumors. Although it was reported that lower serum levels of soluble CD44 v6 were significantly associated with poor prognosis in patients with PDAC, it might not accurately reflect the prognosis of CD44+ in patients with PDAC. 

We have demonstrated that overexpression of CD44 has a statistically significant association with a decreased 5-year overall survival rate in patients with PDAC after surgery. The prognosis of CD44− patients indicated a significantly better survival, wherein the tumor staging between the two groups of patients showed no difference. The primary factor appears to be related to the presence or absence of tumor stem cells in their PDAC. It has been reported that, in several cancers, only a minority of the cancer cells are able to initiate the formation of new tumors. These cancer-initiation cells are known as CSCs, and several studies have demonstrated that the CD44 adhesion molecule is expressed specifically in these cells. It has also been reported that CD44+ cancer cells contain CSC-like properties and can initiate in vivo tumor formation. Recent studies have shown that CSCs exhibited increased resistance to drug and radiation therapies. In the present study, we demonstrated that CD44 overexpression is associated with poor prognosis, which suggests that cancer cells strongly expressing CD44 have qualities related to CSCs. These results support the hypothesis that increased cell adhesion supports the growth of cancer cells.

Cancer stem-like cells were found to be expanded during the acquisition of gemcitabine resistance. After exposure to high-dose gemcitabine, CD44-positive cells re-proliferated and reconstituted the resistant cell population. Among the ABC transporters, the expression of AGS was significantly augmented in resistant cells. CD44 is a well-known adhesion molecule and membrane receptor for hyaluronan, and is involved in cell motility and metastases. The gene encoding CD44 generates a variety of isoforms by alternative splicing, which predominantly affects the extracellular membrane-proximal structure of CD44 proteins. The expression of CD44 variants showed a significant correlation with poor prognosis in colon and pancreatic cancer. Recently, CD44 has been evaluated as a CSC marker in solid tumors. In fact, CD44 alone served as a CSC marker in head and neck carcinoma. The proportion of CD44+ subpopulation in primary tumors varies from 0.1% to 42%. However, in other tumors, multiple cell surface markers have been used to purify CSCs, and CD44 has been used as one of the CSC markers. In pancreatic cancer, although CD44+ cells themselves were more tumorigenic than CD44− cells, CD44+/CD24+/ESA+ cells had more stem cell-like characteristics than CD44−/CD24−/ESA− cells. When patients’ samples were sorted using these markers, 2–9% of cells expressed CD44, whereas only 0.2–0.8% of cells were CD44+/CD24+/ESA+. Although CD44 itself is not sufficient to define all the phenotypes of CSCs, it appears obvious that CD44 is one of the important CSC markers in pancreatic cancer. Based on the CSC model of drug resistance, after treatment with high-dose gemcitabine surviving CSCs might differentiate into progenitor cells, and these cells would proliferate and differentiate to generate recurrent tumors [5]. These data show that CD44+ cells are primarily responsible for this process. There is no consensus regarding the proportion of CSCs in primary or recurrent tumors; however, it has been accepted that CSCs may represent a small proportion of heterogeneous tumor cells. Our data also revealed that CSCs were confined to a small subset in recurrent and resistant cell populations. CSCs are naturally resistant to chemotherapy through their properties of quiescence, capacity for DNA repair, and ABC transporter expression.

Among the ABC transporters, ABCG2 and ABCB1 have been well-investigated in stem cells. It has been reported that ABCG2 is exclusively expressed in stem cells, in which the expression is easily turned off in the majority of committed progenitor and mature blood cells [19]. In the present study, the expression of ABCG2 was found to be augmented during the acquisition of multidrug resistance. We further observed that the expression of ABCB1 was gradually increased, along with the proliferation of CD44+ cells. Paraffin samples of pancreatic cancers were stained with anti-CD44 antibody. CD44 was stained in the membrane of cancer cells, and its expression correlated with the tumor grade. Patients with CD44-expressed pancreatic cancers showed poor prognosis. 

Nanog formed a complex with STAT-3 activated MDR1 expression, and HA-CD44 interaction also activated MDR1 through ankyrin, resulting in multidrug resistance. This explains why ABCB1 was strongly augmented in resistant cells as CD44+ cells proliferated. Due to these reasons, the inhibitors of ABC transporters have been investigated in the treatment of cancers, and our data also suggest that they could reverse chemoresistance in pancreatic cancer.

In this study, we have demonstrated that CD44−–targeted therapy is a possible option for reversing chemoresistance of pancreatic cancer cells. In acute myeloid leukemia, the administration of a monoclonal antibody against CD44 markedly reduced the repopulation of leukemic stem cells in vivo; however, this effect was due to the inhibition of proper homing of leukemic stem cells to microenvironmental niches. Based on our data showing that CD44+ cells reconstitute the resistant cell population, CD44 could be used as a therapeutic target to overcome drug resistance and cure the disease. Although we have demonstrated the role of CD44 in this study, a recent report identified CD133 as a CSC marker in pancreatic cancer. That study reported that CD133^+^ cells were resistant to gemcitabine. This finding suggests that not just one marker, but a set of surface markers would denote pancreatic CSCs. In conclusion, cancer stem-like cells play a pivotal role in acquiring multidrug resistance in pancreatic cancer; CD44+ cells in particular, which repopulate after chemotherapy, were responsible for the chemoresistance mediated by ABCB1. In therapeutic applications, targeted therapy against CD44 or ABC transporter inhibitors could be used to overcome drug resistance and may be beneficial in the treatment of pancreatic cancer.

## 4. Materials and Methods

### 4.1. Human Pancreatic Cancer Cell Lines and Culture Conditions

The human pancreatic cancer cell line Capan-1 was purchased from the American Type Culture Collection (ATCC, Manassas, VA, USA). MIA PaCa-2, PANC-1, and BxPC-3 cells were purchased from the Bioresource Collection and Research Center (BCRC, Hsinchu, Taiwan). AGS, a human gastric cancer cell line, was purchased from BCRC. All cells were maintained in DMEM/F-12 medium supplemented with penicillin (100 U/mL) and streptomycin (100 μg/mL) (Thermo Scientific, Logan, UT, USA). Capan-1 cells were supplemented with 15% fetal bovine serum (FBS) (Gibco, Grand Island, NY, USA). MIA PaCa-2 cells were supplemented with 10% FBS and 2.5% horse serum (Gibco, Grand Island, NY, USA). PANC-1, BxPC-3, and AGS cells were supplemented with 10% FBS. All cells were incubated at 37 °C in a humidified atmosphere with 5% CO_2_.

### 4.2. Identification and Characterization of Human Pancreatic CSCs

We first identified pancreatic CSCs from human pancreatic cancer cell lines. To determine whether CSCs possess a chemoresistant phenotype, we examined the sensitivity of unselected tumor cells, sorted CSCs, and tumor spheroid cells to fluorouracil (5-FU), gemcitabine (GEM), and cisplatin (Sigma-Aldrich, Taufkirchen, Germany) under stem cell conditions or differentiating conditions.

Culture conditions favoring the proliferation of undifferentiated cells were selected for the propagation of the CSC fraction of tumor cells. To avoid adhesion and subsequent differentiation of CSCs, the cells were dissociated from culture plates and placed under stem cell conditions. Cells were grown in serum-free DMEM/F12 medium supplemented with 5 μg/mL insulin, 20 ng/mL human recombinant epidermal growth factor (EGF), 10 ng/mL basic fibroblast growth factor (bFGF), and 0.4% bovine serum albumin (BSA) (Sigma-Aldrich, Taufkirchen, Germany), followed by culturing in ultra-low attachment plates and subsequent organization into spheres. Cells grown under these conditions as non-adherent spherical clusters of cells are generally termed as “spheres” or “mammospheres”. After 10–15 days of plating, the formation of spheres can be observed in the culture. Primary spheres can be enzymatically dissociated into single cells, which in turn give rise to secondary spheres; this procedure can be repeated, which results in an extensive amplification of cell numbers.

CSCs were identified using the surface markers CD44, CD24, ESA (Ep-CAM), and CD133. CSCs were isolated by magnetic bead sorting using the MACS system (Miltenyi Biotech, Bergisch Gladbach, Germany). For this purpose, cultured tumor spheres and tumor cells were trypsinized, washed, and resuspended in PBS. Cells were then incubated with a surface marker monoclonal antibody labeled with MicroBeads (Miltenyi Biotech) for 30 min at 4 °C, and cells were enriched using a MACS magnet and MS columns (Miltenyi Biotech). All MACS procedures were performed according to the manufacturer’s instructions. The purity of isolated cells was determined by standard flow cytometry analysis using an APC-labeled antibody against a human CSC surface marker.

We also isolated CSCs by fluorescence-activated cell sorting (FACS) using the markers CD44, CD24, ESA, and CD133, either individually or in combination. Dissociated cells were counted and transferred to a 5-mL tube, washed twice with HBSS containing 2% heat-inactivated FBS, and resuspended in HBSS containing 2% FBS at a concentration of 10^6^/100 µL. Sandoglobulin solution (1 mg/mL) was then added to the sample at a dilution of 1:20, and the sample was incubated on ice for 20 min. Then, the sample was washed twice with HBSS contained 2% FBS and resuspended in the same medium. Antibodies were added and the sample was incubated for 20 min on ice, followed by washing twice with HBSS containing 2% FBS. When required, a secondary antibody was added by resuspending the cells in HBSS containing 2% FBS, followed by incubation for 20 min. After another washing, cells were resuspended in HBSS containing 2% FBS and 4′,6-diamidino-2-phenylindole (DAPI; 1 µg/mL final concentration). The antibodies used were anti-CD44 allophycocyanin (APC), anti-CD24 phycoerythrin (PE), anti–ESA-FITC, and anti-CD133 APC, each at a dilution of 1:40. Dead cells were eliminated using the viability dye DAPI. Flow cytometry was carried out using a FACSAria (BD Bioscience, San Jose, CA, USA). Side scatter and forward scatter profiles were used to eliminate cell doublets. Cells were routinely sorted twice and reanalyzed for purity.

### 4.3. Non-Adherent Sphere Assays for CSCs

The stem cell activity of isolated pancreatic CSCs was evaluated by a useful in vitro assay, the “sphere” assay. The ability of cells to form colonies in spherical aggregates in non-adherent culture conditions is reflective of cells with a self-renewal capacity. Each cell was diluted to a density of 10^3^ cells/mL with serum-free medium (SFM), i.e., DMEM/F12 supplemented with 10 ng/mL fibroblast growth factor, 20 ng/mL epidermal growth factor, and 2.75 ng/mL selenium (insulin-transferrin-selenium solution) (Invitrogen, Carlsbad, CA, USA). Then, 100 µL of diluted cell suspension was seeded into each well of a 96-well low attached plate at a density of 10^2^ cells/well. At day 7, 100 µL of SFM was added to each well. At day 15, spheres larger than 50 µm were counted under a microscope.

### 4.4. Colony Formation Assay

To determine the colonogenicity of CSCs in vitro, we compared their clonal ability to that of parental cancer cells using a soft agar colony formation assay. In this assay, cells may be passaged in double-layer soft agar or embedded in Matrigel, a substitute for the basement membrane. Each colony-forming assay represents a read-out for progenitor cell activity. Colony-forming efficiency was determined using a double-layer soft agar method. A total of 10^4^ cells were plated onto 0.35% agar over a layer of 0.5% agar containing DMEM and 10% FBS in 6-well plates. Cells were incubated for 21–28 days in a CO_2_ incubator, and colonies larger than 50 µm were counted under a microscope.

### 4.5. Implantation of Sorted CSCs into NOD/SCID Mice

We evaluated the in vivo tumorigenic potential of isolated pancreatic CSCs. Dilutional tumor propagation assays and orthotopic transplantation of sorted CSCs into immunocompromised mice are gold standards for evaluating the presence of CSCs. Cells were isolated by flow cytometry based on the expression of the surface marker, and their ability to form tumors was analyzed. Sorted cells were washed with serum-free HBSS and suspended in a serum-free RPMI/Matrigel mixture (1:1 volume), followed by sc injection into the flank of NOD/SCID mice at concentrations of 100, 500, 10^3^, and 10^4^ cell per injection. Mice were examined for tumor formation using digital calipers and by subsequent autopsy. The analysis was completed 16 weeks after the injection.

In orthotopic transplantation experiments, mice were anesthetized using an ip injection of 100 mg/kg ketamine and 5 mg/kg xylazine, followed by median laparotomy. Then, either 1000 or 5000 sorted cells (versus-negative) resuspended in PBS in a volume of 100 µL were injected into the tail of the pancreas using a 30-gauge needle. Animals were subjected to autopsy at 28 days after cell implantation, and the tumor growth was assessed. Tissues were fixed in formaldehyde and examined histologically.

### 4.6. Immunocytochemical Staining

Immunostaining of all cell types was performed as described, except the cell surface labeling of ABCG2. For labeling of pluripotent and differentiation markers, the following primary antibodies were used: Oct4 (SantaCruz, Dallas, TX, USA), Nanog, SSEA4, and PODXL (R&D Systems, Minneapolis, MN, USA), cardiac troponin-I (Sigma, St. Louis, MO, USA), Nestin (Abcam, Cambridge, UK), b-III tubulin (R&D Systems), and CD-44-FITC (BD Biosciences). The ABC transporters were analyzed using the same antibodies used for flow cytometry. Hoechst33342 (Invitrogen) was used for nuclear staining. The stained samples were examined by an Olympus FV500-IX confocal laser scanning microscope. Additional details can be found in the supporting information.

### 4.7. Immunofluorescence Staining

Unselected tumor cells, sorted CSCs, and tumor spheroid cells were cultured in 24-well plates and fixed in a solution of 4% paraformaldehyde/0.2% Triton in PBS for 30 min at 4 °C. Cells were then washed in PBS, treated with PBS/5% milk for 60 min at room temperature, and then stained with the following primary antibodies at 4 °C overnight: anti-CD44 allophycocyanin (APC), anti-CD24 phycoerythrin (PE), anti-ESA-FITC, and anti-CD133 APC. Cells were counterstained with phalloidin-conjugated FITC or Texas Red. Nuclei were stained with DAPI. Cells were then washed twice as described above and observed under the fluorescence microscope. Isotypes and non-probed cells were used as controls.

### 4.8. Establishment of Gemcitabine-Resistant Pancreatic Cancer Cells

Gemcitabine-resistant pancreatic cancer cells were established using escalating doses of gemcitabine serially in BxPC-3 and Capan-1 cells. Initially, cells were cultured for 72 h with IC_50_ of gemcitabine with a defined drug-free interval. As the cells began adapting to the dose, the concentration of gemcitabine was doubled serially. Finally, after recovering the cells from 10 μM gemcitabine treatment, 100 μM of the drug was added into the medium to remove most of the cell population. The experiments described below were performed using the recovered cells.

### 4.9. Chemotherapy Sensitivity Assays

To determine whether CSCs possess a chemoresistant phenotype, the sensitivity of unselected tumor cells, sorted CSCs, and tumor spheroid cells to 5-FU, gemcitabine (GEM), and cisplatin were analyzed under stem cell conditions or differentiating conditions. Culture conditions were optimized for all cell lines. Cells were plated in 96-well plates at day 1 at different densities based on their doubling times (3000–15,000 cells per well). The optimal plating number was the highest number of cells that enabled log-linear growth for 4 days. Log-linear growth or exponential growth is the phase of growth during which each descendant of the parental cell divides as well. Cells were dissociated (trypsinization and filtering through 40-µm cell strainers) and seeded at optimal density per well in 100 µL medium. After 24 h, an additional 100 µL of medium containing various concentrations of antitumor agents, ranging from 0.01 to 100 μg/mL (5-FU), 0.0001 to 1 μg/mL (GEM), and 0.1 to 50 μg/mL (cisplatin), was added and incubated for another 72 h. Relative cell numbers were determined by standard 3-(4,5-dimethylthiazol-2-yl)-2,5-diphenyltetrazolium bromide (MTT) assays. Dose–response experiments were performed in duplicate. Doses required for 50% growth inhibition (IC_50_ values) were determined using Prism 4 (GraphPad Software) as mean ± SD.

### 4.10. Investigation of the Role of ABC Transporters in CSCs

Although the mechanisms of multidrug resistance in CSCs are poorly understood, ABC transporters such as ABCG2, ABCB1, and ABCC1 were believed to be responsible. Therefore, mRNA expression levels of ABCG2, ABCB1, and ABCC1 were measured by real-time RT-PCR. 

Furthermore, we evaluated rhodamine 123 intracellular uptake assay to determine the functional activity of ABCB1 in CSCs. A total of 500,000 cells were harvested after trypsinization and resuspended in PBS. Rhodamine 123 was added at a final concentration of 1 μM, and the cells were incubated in a water bath at 37 °C in the dark. After washing twice with ice-cold PBS, flow cytometry was carried out using FACSAria (BD Biosciences, San Jose, CA, USA).

The inhibition of ABC transporters may reverse the chemoresistance of CSCs. To test this hypothesis, clonogenic assays using antitumor drugs plus verapamil were conducted. Verapamil has been demonstrated to be an inhibitor of ABC transporters. Briefly, 1000 cells were seeded in 6-well plates and incubated for 72 h. Then, the cells were incubated with various doses of verapamil and antitumor drugs for 72 h. After incubation for another 7 days, colonies consisting of more than 32 cells were counted under a microscope.

## Figures and Tables

**Figure 1 ijms-24-07331-f001:**
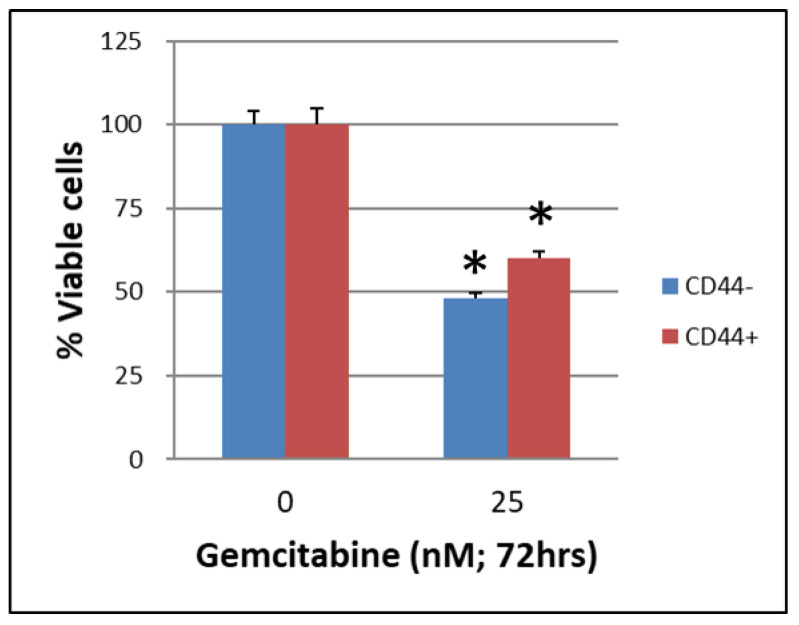
The survival of cancer stem cells CD44+/CD44− of BxPC-3 PDAC cells treated with or without gemcitabine (25 nM, 72 h) is shown. The CD44+ had better survival than CD44−, * < 0.05. (Mean ± SD).

**Figure 2 ijms-24-07331-f002:**
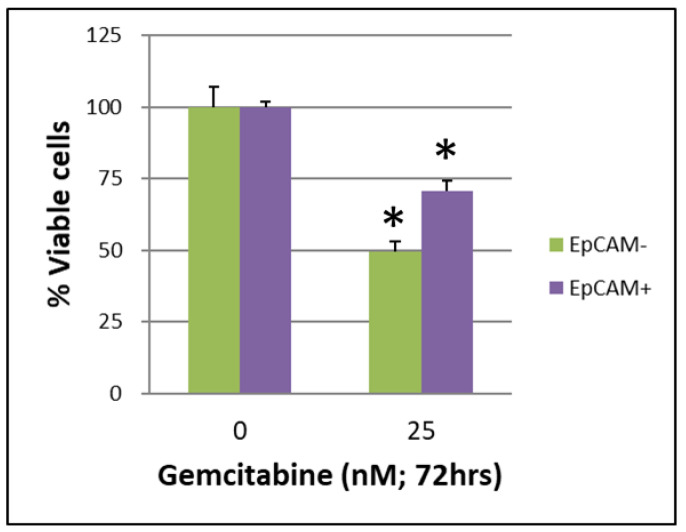
The survival of cancer stem cells EpCAM+/EpCAM− of BxPC-3 PDAC cells treated with or without gemcitabine (25 nM, 72 h) is shown. The EpCAM+ had better survival than EpCAM−, * < 0.05. (Mean ± SD).

**Figure 3 ijms-24-07331-f003:**
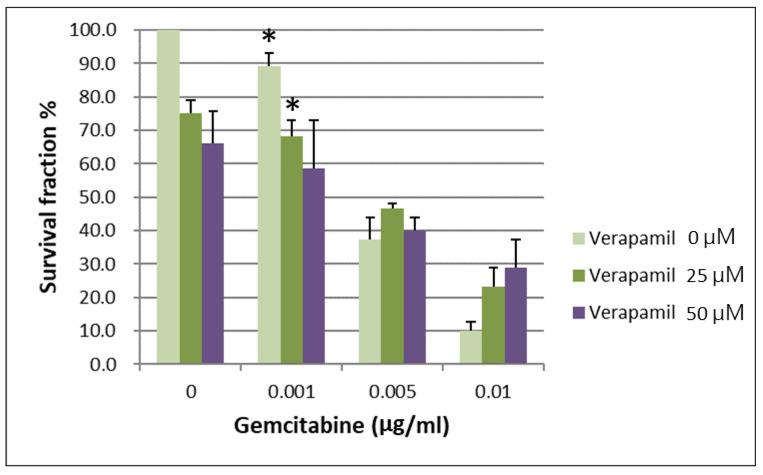
The survival of parental BxPC-3 PDAC cells treated with different concentrations of verapamil and gemcitabine is shown. Verapamil 25 and 50 µM exerted the best chemotherapeutic effects when gemcitabine concentration of 0.001 µg/mL was used, * < 0.05. (Mean ± SD).

**Figure 4 ijms-24-07331-f004:**
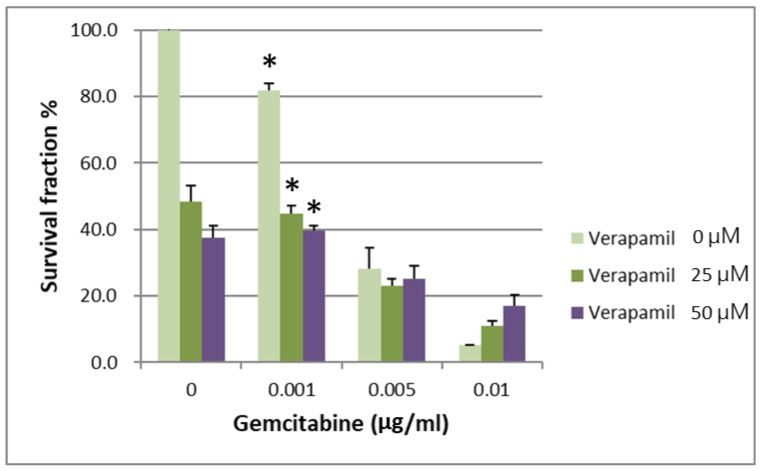
The survival of CD44+ cancer stem cells of BxPC-3 PDAC cells treated with different concentrations of verapamil and gemcitabine is shown. Verapamil 25 and 50 µM exerted the best chemotherapeutic effects when gemcitabine concentration of 0.001 µg/mL was used, * < 0.05. (Mean ± SD).

**Figure 5 ijms-24-07331-f005:**
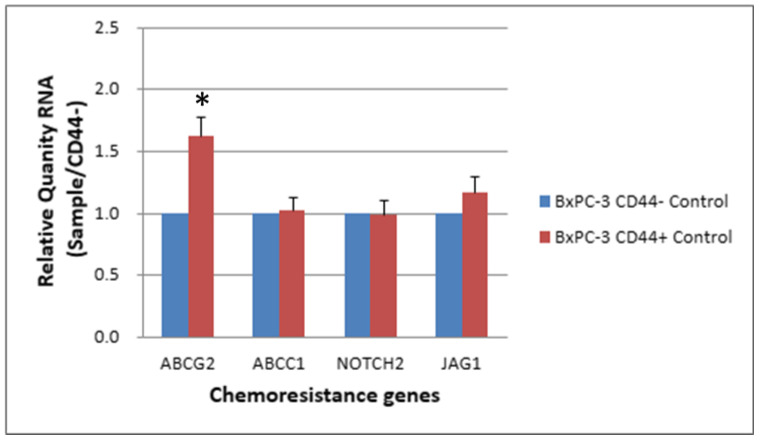
The expression level of 4 chemoresistance genes for non-drug treatment CD44− cancer stem cells and CD44−BxPC-3 PDAC cells is shown. The ABCG2 gene had significantly higher expression in CD44+ than CD44−, * < 0.05. (Mean ± SD).

**Figure 6 ijms-24-07331-f006:**
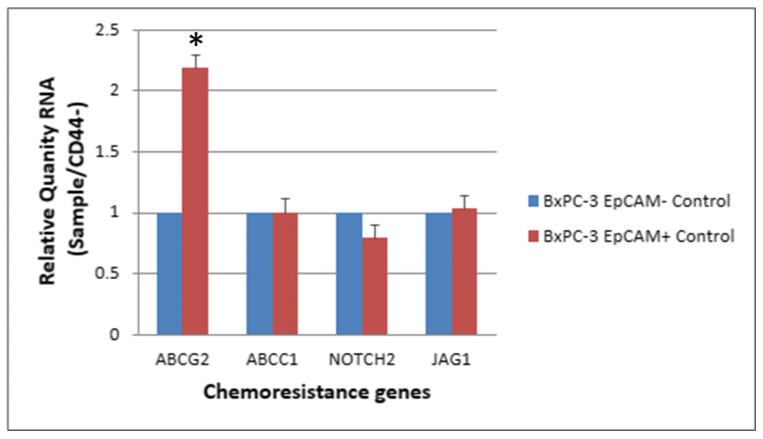
The expression level of 4 chemoresistance genes for non-drug treatment EpCAM+ cancer stem cells and EpCAM− BxPC-3 PDAC cells is shown. The ABCG2 gene had significantly higher expression in EpCAM+ than EpCAM−, * < 0.05. (Mean ± SD).

**Figure 7 ijms-24-07331-f007:**
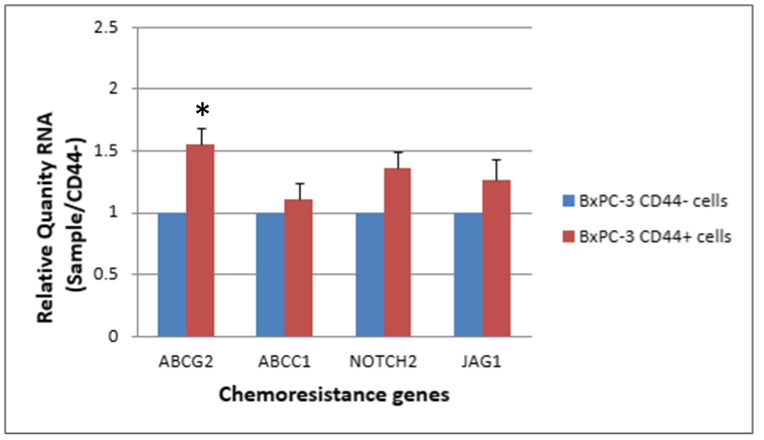
The response of 4 chemoresistance genes for CD44+ cancer stem cells and CD44−BxPC-3 PDAC cells treated with gemcitabine (25 nM, 48 h) is shown. The ABCG2 gene had significantly higher expression in CD44+ than CD44−, * < 0.05. (Mean ± SD).

**Figure 8 ijms-24-07331-f008:**
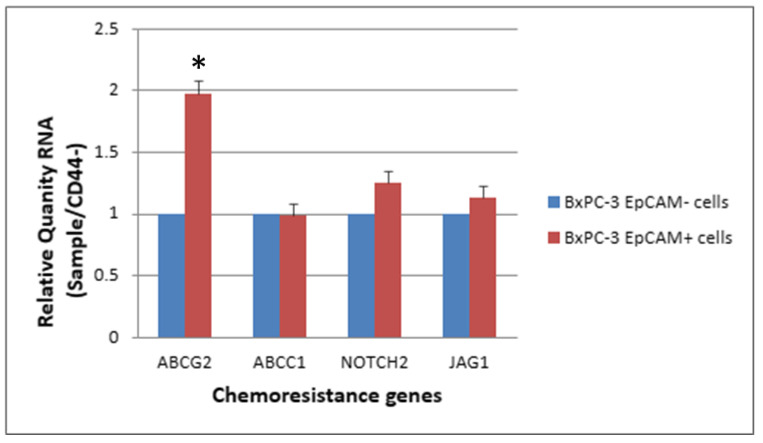
The response of 4 chemoresistance genes for EpCAM+ cancer stem cells and EpCAM− BxPC-3 PDAC cells treated with gemcitabine 25 nM, 48 h is shown. The ABCG2 gene had significantly higher expression in EpCAM+ than EpCAM−, * < 0.05. (Mean ± SD).

**Table 1 ijms-24-07331-t001:** IC50 values for gemcitabine in respective BxPC-3 CSCs cell fractions.

Cell Fractions	Gemcitabine	
	IC50 (nM)	% Viability at 25 nM
CD44−	21	48.2
CD44+	30	60.2
EpCAM−	23	49.4
EpCAM+	39	70.5

**Table 2 ijms-24-07331-t002:** ABC transporters with expression altered by >1.5 fold in CD44+. cells compared with CD44− cells (RQ: Relative Quantity RNA).

BxPC-3 Cell				BxPC-3 Cell + Gemcitabine (20 nM, 48 h)			
1st Test		2nd Test		1st Test		2nd Test	
CD44+/CD44−		CD44+/CD44−		CD44+/CD44−		CD44+/CD44−	
Gene	RQ	Gene	RQ	Gene	RQ	Gene	RQ
ABCB1	1.81	ABCB5	2.04	ABCC9	4.22	ABCA3	1.67
ABCC9	1.45	ABCC12	2.36	ABCG5	1.89	ABCC9	7.25
ABCD4	1.36	ABCD3	1.57			ABCD2	1.66
		ABCD4	1.68			ABCG8	2.37
		ABCG4	1.93				

## Data Availability

The data will be opened after publication.

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
