# Peer review of "Identification and Clinical Significance of Pancreatic Cancer Stem Cells and Their Chemotherapeutic Drug Resistance"

_ijms, 2023, doi:10.3390/ijms24087331_

Round 1
Reviewer 1 Report
Comments
Authors compiled a research manuscript on “Identification and Clinical Significance of Pancreatic Cancer 2 Stem Cells and their Chemotherapeutic Drug Resistance”. Authors compiled this manuscript very well but still it requires substantial revision based on the mentioned points before the final publication given as follows:
1. Authors did not mention the details of identification of chemo resistant genes like how they identified, RT-PCR bands, effect of gene variants etc and man more?
2. Apart form graphs, no other data is provided to justify the results of each study. Please provide the evidence of each study in terms of images and tables.
3. How authors identified the pancreatic CSC from human pancreatic cancer cell lines? Please provide justification.
4. This manuscript overall lack in data presentation. Please improve it.
Reviewer 2 Report
1) Figure 1 and 2 how the 25 nM concentration of gemcitabine was selected? which are the error bars SEM or SD? Identify the statistics differences? p values are not identify in the figure. I don't think that it is a big difference in Figure 1 to conclude a better survival rate. Combine figure 1 and 2.
2) Explain figure 4, which results are from CD44+ cells and which one from CD44-. Include statistics and error bars information SEM or SD?
3) Figure 5, 6, 7 and 8 include error bars, there is information about statistic in the figure title but not in the figures.
4) Which is the difference between figure 5 vs. 7, and figure 6 vs. 8. According to the title is the same experiment. Improve the presentation of your results it is difficult to understand.
Reviewer 3 Report
The manuscript by Kuo et al. discusses the identification and clinical significance of pancreatic cancer stem cells (CSCs) and their chemotherapeutic drug resistance. They demonstrated that cancer stem-like cells play an important role in acquiring multi-drug resistance in pancreatic cancer, and CD44+ cells in particular. This study potentially sheds light on how to overcome chemoresistance and its mechanisms in pancreatic CSCs. The claims are mostly supported, and the conclusions appear to be sound. This reviewer does not have further suggestions beyond the writing styles as following-
- Albeit the introductory paragraphs are comprehensive, it appears to be rather lengthy, and could be potentially cut down to set up a clearer research question and hypothesis.
- The conclusion section is concise, but it could be improved by providing an outlook of the implications of this research from a broader scope.
Round 2
Reviewer 1 Report
No furhter revision is needed
Reviewer 2 Report
They addressed all the comments.